# Effect of the Shape of Styrene–Acrylonitrile Water-Filter Housings on the Destructive Pressure, Crack-Initiation, Propagation Conditions and Fracture Toughness of Styrene–Acrylonitrile

**DOI:** 10.3390/polym12020280

**Published:** 2020-01-31

**Authors:** Borut Zorc, Matija Zorc, Borut Kosec, Aleš Nagode

**Affiliations:** 1Welding Institute Ltd., Ptujska 19, SI-1000 Ljubljana, Slovenia; 2Faculty of Natural Sciences and Engineering, University of Ljubljana, Aškerčeva 12, SI-1000 Ljubljana, Slovenia; matija.zorc@omm.ntf.uni-lj.si (M.Z.); borut.kosec@omm.ntf.uni-lj.si (B.K.); ales.nagode@omm.ntf.uni-lj.si (A.N.)

**Keywords:** styrene–acrylonitrile, geometrical shape, stress concentrator, destructive pressure, fracture toughness

## Abstract

A destructive pressure test of styrene–acrylonitrile (SAN) water-filter housings showed the influence of the shape and specific details of the housings on their critical areas and their destructive pressure. The destructive pressure varies by as much as 37 bar due to different dominant stresses in the individual types of housings. In critical areas of the housings, geometrical stress concentrators generally exist. For this reason, the stress caused by the internal pressure is locally 2.75–3.4 times greater than that expected based on the water pressure, which means that cracks are initiated in these places. However, the bottom of the housings can be in a form such that the maximum stress and the crack originates in its central part without the influence of local stress concentrators. The tensile strength of the SAN is theoretically estimated at 73 N/mm^2^, which is comparable with the literature data. The fracture toughness of the SAN is typically low, theoretically estimated in the range 1.45–3.55 MPa·m^1/2^, and strongly depends on the degree of the wall’s stress-increasing rate or the crack-propagation rate. Therefore, at various crack-propagation rates, the critical crack depths are also different, in the range 100–600 μm. Due to this, the critical thickness for brittle fracture in the SAN is also different; it is ten times greater than the critical crack length. The characteristic of a sub-critical crack, i.e., the mirror zone, is its macroscopically smooth surface, which is microscopically very finely roughened. In the case of a sufficiently slowly growing sub-critical crack, the surface of the mirror zone contains characteristic parabolic markings. The over-critical, sufficiently rapidly growing cracks generally grow mainly in the plane-strain state and only the final thin layer of the remaining wall thickness breaks in the plane-stress state. The over-critical, sufficiently slowly growing cracks grow in the plane-stress state with a strong shear plastic tearing.

## 1. Introduction

In the water-supply system of individual buildings, water filters with cylindrical housings are often installed. Some of the housings have external vertical ribs on the walls to the full height, while others have a smooth wall. The cylindrical housing with external ribs is screwed directly onto the filter head with a thread, and the cylindrical housing without external ribs is indirectly fixed to the filter head with a specially formed nut. The filter housings are made of styrene–acrylonitrile (SAN), the filter head and the specially formed nut are made of polypropylene (PP). In Slovenia, several cases of leakage from water-filter housings have been identified. So far, the known leaks came from housings with the external vertical ribs. In addition, fatigue cracks along the external vertical ribs have been found in already used, but still tight, water-filter housings [1]. Since the cracks always occur in the region of the maximum stresses or locally exceeded material strength, the sharp transition between the ribs and the wall of the housings represents the local elastic stress concentrator *K_t_*, which increases with the water-pressure-induced stresses by more than three times [2]. Two stress-concentrators are known: the theoretical elastic stress-concentrator factor *K_t_* is used for static loads in the Hooke’s law area, and the fatigue stress-concentration factor *K_f_* is used for alternating loads [3,4,5,6,7]. Since the yield strength *R_y_* and the tensile strength *R_m_* of the SAN are practically the same (*R_y_*/*R_m_* ≈ 1), the factor *K_t_* is used until rupture and, therefore, can also be applied during a destructive pressure test [2]. Due to the very small number of cycles, the destructive pressure test with a manual water pump is a special example of low-cycle fatigue, where the step-like increasing of the stress can be assumed to be a continuously increasing stress, since a few cycles do not affect the tensile strength of the material.

The article discusses the influence of the shape, the specific details and the local elastic stress-concentrator factors *K_t_* on the strength of the various cylindrical, volumetrically equal, SAN water-filter housings from the same manufacturer. The macroscopic and microscopic analyses of the cracked surfaces determined the initiation sites and the direction of the crack growth. The influences of the increasing stress rate and the crack-growth rate on the texture of the fractured surfaces were qualitatively assessed. With the equations for the calculation of the stress in the thin-walled pressure vessels, the strength of the SAN was determined, and the corresponding value of the graphically determined *K_t_* factors was also confirmed. With fracture-mechanics equations, the fracture toughness of the SAN and the influences of the crack-growth rate on the fracture toughness were estimated and the mode of the crack propagation was explained.

## 2. Experimental

### 2.1. Material

The water-filter housings were made of SAN. Since the housings were purchased on the market, the mechanical properties and the proportion of acrylonitrile in the SAN are unknown. The only available technical information (technical data from the sale catalogue) is the destructive pressure of the housings, which is *p_d_* ≥ 32 bar.

### 2.2. Testing

The investigations were based on a destructive pressure test. Before the destructive pressure test, detailed visual inspections of the housings with the naked eye and with a 5× magnifying glass were performed. Different un-homogeneities that could affect the destructive pressure or the potential site of the crack initiation were searched for. The purpose was to select the housings for which the macroscopically visible production defects would not have effects on the housings’ destruction. Before the destructive pressure test, all the typical dimensions and geometrical characteristics of the filter housing of both types were measured with a calliper “Garant absolute” and a “Mitutoyo PJ-3000” profile projector with a magnification of 10× (both instruments have a calibration certificate). The geometry and dimensions of the housings are important for the theoretical analysis. The destructive pressure test was carried out with a hand-operated water pump in 2–4 cycles (calibrated water manometers with a range of 0–50 bar and 0–100 bar were used). The crack surfaces were examined with the naked eye, a 5× magnifying glass and with a scanning electron microscope (SEM JEOL JSM-5610, analysis performed at 20 kV on gold-plated samples).

### 2.3. Types and Geometrical Characteristics of the Water-Filter Housings

Three types of SAN water-filter housings with the same volume were tested: the housing with external vertical ribs is a product from the market (Figure 1a); the housing without vertical ribs is a product from the market (Figure 1b); the housing without vertical ribs with an increased bottom thickness is the laboratory sample, where the thickness of the bottom was increased with epoxy resin (Figure 1c).

All the housings are in the form of a cylinder with the same internal diameter and have the shape of a drinking glass. The diameter is, therefore, slightly reduced towards the base of the housings (the upper external diameter is *D_exu_* = 71.6 mm, while at the bottom *D_exb_* = 68.6 mm, and the average diameter is *D_ex_* = 70.0 mm). The housing types differ in the upper part due to different ways of fitting to the filter head. The housing with the eight external ribs is directly threaded to the filter head made of PP. The housing without the external vertical ribs is thread-less and is fixed with a PP nut onto the filter head (see Section 2.4). The housings also differ slightly in the thickness of the wall: the average thickness of the housing with the vertical ribs is *t_w_* = 4.7 mm, and without the vertical ribs is *t_w_* = 4.3 mm. The thickness of the base of the laboratory sample (Figure 1c) was *t_b_* ≈ 24.0 mm. The shape and dimensions of the base of the two types of housings from the market are the same (Figure 1d,e). The only difference is that in the case of the housing with the external vertical ribs, they pass into the base continuously through a curved part (Figure 1a), while in the housings with a smooth external surface the external ribs exist only on the curved part of the base and do not stand out above the external surface of the cylindrical part (Figure 1b–d). The thickness of the wall of the curved transition in both housings is *t_w_* = 4.3 mm (Figure 1e). The transition radius between the ribs and the wall of the housing is *r*_1_ = 0.2 mm (Figure 1f), and the average height of the ribs is *t_r_* = 2.1 mm. In the housing without ribs, the clamping ring is 7.2 mm wide and 9.0 mm high. The transition radius between the clamping ring and the wall of the housing is *r*_2_ = 0.3 mm (Figure 1g). The width of the clamping ring locally increases the wall thickness of the upper part of the housing to *t_wc_* = 7.2 mm. Both radii *r*_1_ and *r*_2_ are very sharp and represent strong local stress concentrators.

Although preliminary investigations confirmed that the thin surface lines in the housings with vertical ribs do not have an effect on the initiation and spread of the crack [1,2], various other macroscopically visible production defects that could affect the destructive pressure and the crack-initiation site were searched for. The base of the housings with the vertical ribs was free of macroscopically visible defects. All the housings without the ribs contained a rough, non-transparent surface depression in the external central area of the base (Figure 1e, arrow). This could have a decisive effect on the strength of the filter housing in the case of sufficiently thinned walls due to the effect of the depression and the roughness as a stress concentrator. Because the base is also the most critical part of the housing in the case when the stress concentrators in the base wall do not exist [8,9], the housings with the smallest defect like this were selected for the destructive pressure test.

### 2.4. Samples for the Destructive Pressure Test

To determine the strength of the water-filter housings, in addition to the actual samples from the market, laboratory samples were also produced. In the laboratory samples, the head and the nut were made of steel. Depending on the present variables, the destructive pressure test was carried out on four different series of samples:-A: the housing with the external ribs is directly threaded to the head of PP (Figure 2a),-B: the housing without the ribs is fixed to the PP head with a PP nut (Figure 2b),-C: the housing without the ribs is fixed to the steel head with a steel ring (Figure 2c),-D: the housing without the ribs but a thicker base is fixed to the steel head with a steel ring (Figure 2c).

Series A and B are samples from the market; series C and D are the laboratory samples. In each series two pieces of the sample were tested.

## 3. Results and Discussion

### 3.1. Destructive Pressure Test

The destructive pressure test was carried out with a manual water pump, while simultaneously recording the manometer with a camera. The lowest measured destructive pressure *p_d_* for the different series of test samples was:-the housing broke at the pressure of *p_d_* = 34 bar,-B: the PP nut broke at the pressure *p_d_* = 48 bar, the housing remained undamaged,-C: the housing broke at the pressure of *p_d_* = 50 bar,-D: the housing broke at the pressure of *p_d_* = 71 bar.


Both the pieces from each series had a higher destructive pressure than stated in the technical data and the destructive pressure of the two pieces differed by Δ*p* ≤ 1 bar. The results show that the destructive pressures and the critical areas of the water-filter housings are very dependent on the shape of the housing, the method of fitting the housing to the filter head and on the material of the fitting system. The water filters from the market have very different strengths. The housing with the external vertical ribs has, despite its thicker wall, the lowest destructive pressure due to the longitudinal stress concentrator at the ribs. Namely, it is *p_d_* = 14 bar lower than for the filter without the vertical ribs. The laboratory samples showed that the housings without the ribs are stronger than the PP nut by *p_d_* = 2 bar, and compared with the housing having the vertical ribs, they are stronger by *p_d_* = 16 bar. The housing without the ribs but with a sufficiently thick base is *p_d_* = 21 bar stronger than the commercially available housing without the ribs and twice as strong (by *p_d_* = 37 bar) than the housing with the ribs.

### 3.2. Analysis of the Fractures

To determine the critical area of the crack-initiation and the crack-propagation direction in the different series of samples, the individual broken pieces were stuck together with transparent adhesive tape. The analysis showed that the different series of samples have different critical areas of crack-initiation, as well as the specific and unique subsequent growth of the cracks, which is also visible from the different textures of the crack surfaces.

It is very important that the two pieces from each series have a higher destructive pressure than stated in the technical data with a very small difference in the destructive pressure, the same crack initiation point, a very similar growth direction and a similar splitting of the cracks. Therefore, just two pieces of each series showed the reproducibility of the results with very little deviation. This indicates stable product properties and the credibility of the results.

#### 3.2.1. Visual Analysis of the Cracks

##### (a) Sample of Series A

The shape of the housing has been studied in detail in [2]; however, here we briefly summarize the findings. The critical area of the housing with the external vertical ribs, which is directly threaded onto the housing of PP, is a sharp transition *r*_1_ = 0.2 mm between the rib and the wall on the outside of the upper part of the housing. This is indicated by the macroscopically visible Wallner lines (Figure 3a). The primary vertical crack spreads along the rib to the bottom and upwards to the threaded region (Figure 3a). It is characterized by a smooth initial area and a lot of short secondary cracks (they grow in the downwards direction from the primary crack with an orientation of approximately 45° relative to the basic crack) and, in general, the fracture surface is rough and non-transparent with a white shade. The downwards-growing primary crack at the rib enters the base and branches in its central part into several secondary cracks, which generally stop in the base or at its edges (Figure 3b). The cracks with sufficient energy can pass the base and continue to grow vertically along the housing upwards, without the external ribs influencing their propagation (Figure 3c).

The upwards-growing primary cracks in the screw groove split into two oppositely growing horizontal cracks, which grow in the threaded grooves. On the rear side of the housing (the front side is that with the primary crack), the horizontally growing cracks join together, and the growth of these new cracks is downwards, to the base (Figure 3c). It is known that shock waves travel in front of each crack, the speed of which is greater than the crack’s growth speed [10,11]. If they encounter a free surface, they reflect back, and in the crack tip this increases the stress-intensity factor *K_I_*. Due to the mutual interaction of the shock waves of both oppositely growing horizontal cracks with the shock waves of the cracks that have passed the base of the housing and due to the reflection from the free surfaces, the stress-intensity factor at the tip of the one newly formed downwards-growing crack greatly increased. Therefore, this crack is immediately branched into several secondary cracks, which extend along the housing towards the base without the influence of the external vertical ribs on its growth direction (Figure 3c). Some of these cracks are joined together on the wall of the housing (Figure 3c) or on the edge of the base with oppositely growing cracks that have been caused by the branching of the primary crack in the middle of the base. This results in the fragmentation of the filter housing into several pieces, in which the threaded part of the housing is most often torn off in the form of a ring. The surface of the cracks on the back side of the housing (rapidly growing cracks) is much smoother and more transparent compared with the primary crack. Due to the initiation of the primary crack on the external surface of the housing, the primary crack grow through the wall first from the external to the internal surface, but as soon as conditions that are characteristic of a pressure vessel are created, its growth through the wall changed direction from the internal to the external surface of the housing. Such growth is a logical consequence of the higher stress-intensity factor on the internal surface of the housing due to the additional water-pressurized crack opening. It is also evident that the cracks grow much faster along the wall than through the wall (Figure 3c).

##### (b) Sample of Series B

The critical area of the housing without the ribs, which is fixed on the filter head with the nut (both made of PP), is the nut. The fracture exists along the threaded groove of the nut, which divides the nut into two rings. Since the SAN housing was not damaged, we did not investigate this case.

##### (c) Samples of Series C

The critical area of the housing without the ribs, which was fixed to the steel head by a steel ring, is its base. The base of the housing was broken into several fragments with the shape of a circular sector (Figure 4a,b). The glued fragments show the crack initiation in the central, thickened part of the base (Figure 4b), while the macroscopically visible Wallner lines indicate that the crack initiated on the external surface in the marginal area of the thickened cylindrical part (Figure 4c). The initial crack branched into four major energy-comparable cracks with a mutual orientation of approximately 90°, which is characteristic of a burst due to internal pressure [7]. These cracks are already branched into several secondary cracks at the very beginning (Figure 4b). All the cracks are radial spreading towards the base and they pass into the wall of the housing. The main cracks grow upwards to the upper edge of the housing (Figure 4d), but the secondary cracks generally stopped at the half-height of the housing. It seems that the short outer ribs in the curved part do not affect the initiation and growth of the cracks (Figure 4b). Interestingly, only the one upwardly growing crack is severely split (Figure 4d), although all are generally of the same length. This is due to the increased stress-intensity factor at the tip of this crack. There are probably two reasons for this. One possibility is that this crack had more energy than the rest and therefore released more powerful impact waves, which also means that the reflected waves from the upper free surface of the housing had higher energy. Another possibility is the interference between the original and the reflected waves, their summing and the formation of the stronger return waves.

The surfaces of all the radial or vertical cracks are more or less rough and non-transparent, with only a slightly white shade. The highest degree of whiteness is on the surfaces of the vertically growing cracks as a result of the partial shear tearing due to a slower growth. The fracture in the curved part of the base and the formation of circular sector’s fragments is the consequence of horizontal cracks that are formed later than the radial cracks (if it were otherwise, the radial cracks could not reach the top of the housing or would not even pass into the housing from the base). Additional evidence also comes from the way of the horizontal cracks spread in the curved part of the individual circular sector’s fragments. It is clear that the fracture in the curved part consists of the two opposing cracks that grew from the adjacent radial cracks and joined together in the middle of the circular sector’s fragments (Figure 4e). Together with the initial site this proves that the maximum stress is in the middle of the base. However, the stress in the curved part cannot be much smaller than in the central part of the base, otherwise the tearing of practically the whole base would not occur. After the initiation and the final achieved length of the radial-vertical cracks in the wall of the housing, in the final stage of the bursting, the stress in the curved part of the base became a priority. This resulted in the rapid growth of two opposite cracks between the radial cracks in the curved part. This is why the surface of these cracks is smoother and more transparent compared with the others. The cracks grow from the internal to the external surface of the wall and the growth along the wall is much faster than through the wall (Figure 4e). Due to such a growth of horizontal cracks, a bending moment caused by the water pressure is generated in the final phase of bursting the housing, which pushed and broke the circular sector’s fragments of the base from the housing.

##### (d) Sample of Series D

The critical area of the housing without the ribs and with the thicker base, which was fixed on the steel head with a steel ring, is the sharp transition radius *r*_3_ = 0.3 mm between the upper clamping ring and the wall on the external side of the housing, where the crack also initiates. This is indicated by the macroscopically visible Wallner lines (Figure 5a). As a result, the water pressure tears off the clamping ring at this line from the remaining housing in one piece (Figure 5b). The direction of the short secondary cracks that grow from the main cracks with an orientation of approximately 45°, show two oppositely growing horizontal cracks with the same speed that joined on the opposite side of the housing (Figure 5c). At this point, a small triangular fragment was pulled out from the housing due to the junction of oppositely growing cracks and the high-pressure operation. Due to the slow growth of both cracks, the impact waves were too small to initiate a new, downwardly growing crack at this site. In this series of samples, two vertical cracks were formed at a distance of 15–20 mm from the initial site, which grew with approximately same speed in the downwards direction (Figure 5d). Obviously, in these two places, the effect of the first oblique, short, secondary cracks and shock waves has an influence (until this point the secondary cracks do not exist), and due to this the tangential stress also starts to influence, but it did not prevail over the axial stress because of the stress concentrator between the clamping ring and the wall. The proof of this is a tear-off of the clamping ring in one piece due to the axial stress. However, the vertical downwards-growing cracks began to split at the very beginning (Figure 5d), indicating the effect of the reflected waves from the base on increasing the stress-intensity factor at their peaks. With their propagation, the branching was intensified. Interestingly, both cracks constantly grow vertically downwards at approximately the same distance from the place of their formation, splitting away from each other and negligibly in the direction from one to the other (Figure 5d). This is certainly due to the large difference in the surface of the housing between the cracks on the front side (small surface, ≈2800 mm^2^) and the rear side of the housing (large surface, ≈13,000 mm^2^). Consequently, the pressure is more effective for the formation of tangential stresses due to the stronger expansion and inflation of the rear side of the housing. At the base there is an interconnecting lateral crack, which is in an orientation with the surface of the epoxy resin (Figure 5d). Even in a uniform housing with such a thick base, the same condition would arise, as this is a typical example of the spread of the crack along the line of least resistance.

Due to the very large number of short and densely arranged secondary cracks, all the downwards-growing cracks have the appearance of a pine twig or feathers (Figure 5e), which is not typical for the branched cracks in the samples of series A and C (Figure 3c and Figure 4d). A similar, but less pronounced, condition is seen for the primary downwards crack in the housing of series A (Figure 3a). This confirms that all the cracks in the D series samples are slowly growing with a strong shear tearing, leading to a large plastic deformation of the material. This is also proven by a more roughly and completely non-transparent, whitish fracture surface, which is also characteristic of the fracture surface of the torn clamping ring (Figure 5b). Cracks without short secondary cracks have a smoother and slightly more transparent surface. Because of the very densely arranged secondary cracks that are interconnected, the high pressure at the end of the test detached individual smaller fragments from the housing (Figure 5d, points A).

#### 3.2.2. Electron Microscopy of Fracture Surfaces

At smaller magnifications, the initial point of the cracks has generally the same texture in all cases. The initial surface of the cracks appears smooth (Figure 6a,c,d and Figure 7a) and is called a mirror zone. Its depth represents the critical depth of the crack [2,7], when uncontrolled growth of the crack leads to the final break. The depth of the mirror zone *a_mz_* is different depending on the different series of tested samples: in series A it is *a_mz_* ≈ 100 μm, in series C it is *a_mz_* ≈ 150 μm and in series D it is *a_mz_* ≈ 600 μm. The greater is the speed of increasing the load or the growth in velocity of the subcritical crack *v_mz_*, the lower are their critical depths, which shows that *a_mz_* is proportional to *v_mz_*^−1^ (in series C the speed of increasing the load is 1.5 times slower than in series A, and in series D it is 2 times slower than in series A). In the case of series D, there are few micro-unhomogeneities in the initial area, which did not affect the crack initiation (Figure 7a). In the case of series C, it is clear that the stress concentrator on the external surface of the base, despite its proximity, did not affect the crack initiation (Figure 6c).

The very large magnification shows the very fine rough surface of the mirror zones (Figure 6b,e and Figure 7b), which proves the initiation and growth of the cracks in the mirror zone with the pre-formation and tearing of the crazes. The micro-texture of the mirror zone can be very different and is obviously related to the stress-increasing rate in the wall or the growth velocity of the subcritical cracks. This is evident from the comparison of the cracks’ surfaces. The fine roughness is expressed in the form of more or less visible lines (Figure 6b,e and Figure 7b), the orientation of which in the series A and C samples is consistent with the growth front of a subcritical crack.

The surface of the mirror zone of the slowest-growing cracks (series D) differs significantly from the other two. Here, the more visible lines are directed towards the direction of the crack growth, while the lines oriented in the transverse direction are much less pronounced (Figure 7b). In addition to the lines, the so-called parabolic markings are generally also present on the crack surface of the mirror zone (Figure 7c). These contours are due to the junction of the main crack with the secondary micro-cracks [11], which are initiated ahead, near the front of the main crack at the sites of micro-unhomogeneities in the material due to the local stress concentration at these sites (the initials are at the focus of the parabola, Figure 7d). In the very initial field just below the surface of the housing, there are no parabolic markings (Figure 7c, upper part of the image). In this area darker spots with no pronounced boundaries and circular and droplet-like markings exist on the crack surface (Figure 7c). A larger magnification shows that the dark spots are the micro-craters that form due to the circular growing micro-cracks from the initial sites (Figure 7e). Even slightly deeper (approximately 100 μm below the surface), the typically parabolic markings exist on the crack surface, the number and size of which are increasing, and their open sides are in the direction of the main crack growth (Figure 7c).

The depth-conditioned formation of a transition zone with the above-described contours on the crack surface of the mirror zone and the number and the size of the parabolic markings are specific consequence of the constant increasing of the crack-growth velocity of the sub-critical main crack and a change in the growth rate of the crack in a shallow sub-surface area. Round craters with no visible edges are the result of the harmonic growth of the two types of cracks. This leads to an ideal fusion of the front of both cracks in the plane of the main crack. Due to this, the edge is not created, because the surface of the micro-cracks continuously passes into the plane of the main crack. Closed shapes with edges indicate that the main crack has slightly overtaken the growth of the micro-cracks towards the main fracture surface and tears the remaining thin layer of material between them. However, when the growth rate of the main crack is already large enough compared with the growth rate of the micro-cracks, the parabolic markings are created. Since the micro-cracks on its front side cannot reach the plane of the faster main crack (the main crack overtakes the micro-cracks, which grows in the opposite direction), a plastic flow and tearing of the thin intermediate layer occurs in the direction of the main crack growth (Figure 7d). On the rear, open side of the parabolic markings, the front of the micro-cracks ideally merges with the front of the main crack due to the simultaneous growth of both types of cracks in the same direction. The open side of the parabolic markings therefore always shows the direction of faster growth of the micro-cracks and the direction of growth of the main crack. In the parabolic markings the lines are radial directed from the initial point due to the tendency for symmetrical growth of the micro-cracks. On the open rear sides of parabolas, they are typically extended in the direction of faster crack growth (Figure 7d). It is obvious that micro-craters with no visible edges, circular, droplet, and parabolic contours are formed on the fracture surface only in the case of sufficiently slow growth of the primary sub-critical main crack, when micro-unhomogeneities in the material ahead, near the front of the main crack, can be expressed as a local stress concentrator. If the growing velocity of the primary crack is larger than the critical velocity for this phenomenon, micro-unhomogeneities in the material simply does not have time to function effectively as a local stress concentrator in front of the main crack. Because secondary micro-cracks do not initiate, the surface of the mirror zone is only a very fine roughness, which is a characteristic of the samples of series A and C.

The fracture surface of the crack at depth *a* > *a_mz_* is no longer smooth, but its texture is the same in all samples. Close to the mirror zone there are macroscopically and microscopically visible semi-elliptic Wallner lines, and outward divergent hackle lines pointing along the crack-propagation direction are characteristic of this situation (Figure 6a,c,d and Figure 7a). Both types of lines indicate a slow, ellipsoid spread of the cracks in the initial phase of uncontrolled growth. In a later, rapid phase of uncontrolled growth, the hackle lines are curved (Figure 8a), which also indicates a different crack growth rate along the wall thickness of the filter housing. The cracks first grow through the wall in the plane-strain state and finally, pass through the plane-stress state, which is shown by the Wallner lines, which are parallel with the external surface of the wall (Figure 8a,b). In the case of series C, the surface of the transverse crack indicates that the wall area in the plane-stress state consists of the two planes at different levels (Figure 8b). We think that is the result of bending the circular sector’s fragments right before the burst due to the pressure action. The Wallner lines and hackle lines generally intersect each other at an angle of about 90° (Figure 8a,b) and from the mathematical point of view they form a system of a single-parametric family of curves with a family of orthogonal trajectories. In the case of the several differently oriented free surfaces, from which shock waves are reflected back towards the front of the growing crack, the Wallner lines can form a complex network, which can lead to an incorrect estimation of the direction of the crack growth.

The surface texture of the crack that slowly grows by shear tearing with a strong plastic deformation and with a lot of short secondary cracks and the appearance of a pine twig (series D) macroscopically looks like a brittle fracture in metallic materials (Figure 8c). There are hackle lines on the surface, while the Wallner lines are not seen. This is due to the splitting of the crack front, where a large number of differently oriented free surfaces at different levels are present, which diffusely breaks up the returned waves. The microscopic analysis confirmed a very strong shear plastic tearing, which due to the brittle SAN appears as a mixed brittle-ductile fracture (Figure 8d). On the fracture surface strong shear tearing appears due to the large number of secondary micro-cracks. These initiated ahead, close to the front of the main crack on the micro-unhomogeneities of the material, which act as a local stress concentrator (similar to the mirror zone). The micro-cracks grow independently at first, and after some time, they merge with the main crack. The initial independent growth of the micro-cracks appears on the fracture surface as more or less circularly symmetrical rosette shapes with a smooth round initial area in the middle, from which the hackle lines radial diverge (Figure 8d).

### 3.3. Assessment of the Conditions for Crack Initiation

It was reported that a crack in brittle polymers begins with previously formed crazes that occur at a sufficiently large plastic deformation (for SAN: *ε_cr_* > 0.02% [12]) and at a sufficiently high stress *σ_cr_*, which is relative to the yield strength *R_y_* usually *σ_cr_* ≤ 0.5*R_y_* [13,14]. The required stress for the formation of crazes is smaller, as the stress action is longer [15]. This means that during a very rapid increase of the pressure during a burst-pressure test, the upper limit value *σ_cr_* = 0.5*R_y_* is appropriate or in the case of brittle polymers *σ_cr_* = 0.5*R_m_* is taken due to *R_y_* ≈ *R_m_*. The stress for the formation of the crazes or the initiation of the cracks is therefore determined from the strength of a brittle polymer, which in our case is not known and needs to be calculated.

In the literature, the tensile strength of the SAN is in the range *R_m_* = 56–82 N/mm^2^ [16,17,18]. The unknown strength of the SAN of the tested water-filter housings can be calculated from the equations for the stress state in the walls of the cylindrical pressure vessel. These can be thin-walled or thick-walled, but the criterion for determining the type of pressure vessel is slightly different. A less strict criterion says that for a thin-walled pressure vessel the ratio of the outer *D_ex_* to the inner diameter *D_in_* is *D_ex_*/*D_in_* < 1.2 [8,9], while a more rigorous criterion requires a nominal radius of the vessel that is at least 10 times the thickness of the vessel’s wall [19]. Housings, in a less strict criterion, belong to a thin-walled pressure vessel: housing with outer ribs: *D_ex_*/(*D_ex_* − 2*t_w_*) = 70/(70 − 2·4.7) = 70/60.6 = 1.155; housing without outer ribs: *D_ex_*/(*D_ex_* − 2*t_w_*) = 70/(70 − 2·4.3) = 70/61.4 = 1.14. On the basis of this finding, the equations for a thin-walled pressure vessel were used for an assessment of the SAN strength. Due to the complete description of the local stress state in the walls of the filter housing, the original equations were further multiplied by the elastic stress-concentrator factor *K_t_* and divided by 10 because the pressure unit is in bar. If *σ_r_* = 0 is assumed due to the small radial stress through the wall, the tangential *σ_t_* and the axial (longitudinal) *σ_a_* tensile stresses in the wall of the cylindrical part of the filter housing due to the internal pressure are [3,4,7,8,9,19,20]:(1)$$\sigma_{t}=0.05\cdot K_{t}\cdot p\cdot D_{in}^{}/t_{w}^{}$$
(2)$$\sigma_{a}=0.025\cdot K_{t}\cdot p\cdot D_{in}^{}/t_{w}^{}$$

The critical part of the pressure vessel (if it does not have enough large stress concentrators on the wall) is always the transition of the cylindrical part to the bottom or its bottom. In this area the radial, tangential, and axial stresses are generated due to the internal pressure (Figure 9).

The longitudinal (meridian) stresses are the greatest at the contact point of the arcs of the larger and the smaller radii. Therefore, they are decisive for the dimensioning of the bottom of the vessel and are usually always expressed as a multiple of the axial (longitudinal) tensile stresses of the cylindrical part of the vessel [8]. The less convex is the bottom and the smaller is the transition radius of the cylindrical part of the vessel into the base, the greater are the meridian stresses [8]. This means that the maximum stresses exist with a flat bottom, which is also characteristic for the base of the water-filter housings. Additionally, it is necessary to consider the method of fitting the base of the housing, which can be supported or fixed. Regardless of the fitting, radial and tangential stresses are the same in the center of the round bottom.

In the supported flat round bottom, the maximum stress is in the middle of the bottom and is given by the Equation [8,9,20]:(3)$$\sigma_{\max}=\sigma_{r}=\sigma_{t}=0.031\cdot K_{t}\cdot p\cdot D_{in}^{2}/t_{w}^{2}$$

In the fixed, round, flat bottom, the maximum stress is at the edges. The state of the stresses on the edges and in the middle of the fixed, round, flat bottom is as follows:-radial stress at the edges [8,9,20]:
(4)$$\sigma_{\max}=\sigma_{r}=0.01875\cdot K_{t}\cdot p\cdot D_{in}^{2}/t_{w}^{2}$$-tangential stress at the edges [20]:(5)$$\sigma_{t}=0.005625\cdot K_{t}\cdot p\cdot D_{in}^{2}/t_{w}^{2}$$
-stress in the middle of the bottom [20]:
(6)$$\sigma_{r}=\sigma_{t}=0.0122\cdot K_{t}\cdot p\cdot D_{in}^{2}/t_{w}^{2}$$


In Equations (1)–(6), the unit for the stress is N/mm^2^. The symbols in the equations are as follows: *K_t_* is the elastic stress-concentration factor, *D_in_* is the internal diameter of the water-filter housing or the base (mm), *t_w_* is the thickness of the wall or the base in the initiation or break area (mm), and *p* is the pressure in the housing (bar). In the continuation the stress calculations in the water-filter housings for individual series are present, taking into account the elastic stress-concentrator factors, which are determined on the basis of Peterson’s diagrams.

#### 3.3.1. (a) Sample of Series A

The crack initiation showed that the critical location of the housing with vertical ribs is a sharp transition between the rib and the wall on the outside of the housing. The sharp transition with radius *r*_1_ = 0.2 mm is a strong elastic stress concentrator *K_t_*. This was theoretically determined from Peterson’s diagrams, in which *K_t_* is defined on the basis of the geometrical characteristics. It is reasonable to determine *K_t_* from all the appropriate diagrams and then calculate its mean value, which is used in stress calculations. To estimate the stress concentrator along the vertical rib, a diagram with a one-sided reduction of the pressure vessel’s wall is appropriate (chart 3.6 in [5]). The ratio between the different wall thicknesses is *H*/*h* = (*t_w_* + *t_r_*)/*t_w_* = (4.7 + 2.1)/4.7 ≈ 1.45, while the ratio between the transition radius and the thinner wall is *r*/*h* = *r*_1_/*t_w_* = 0.2/4.7 = 0.0425. With the extension curve for *H*/*h* = 1.5 to the lower values of *r*/*h*, the value *K_t_*_1_* ≈ 3.1 can be obtained. Newer studies suggest that the *K_t_* values from Peterson’s diagrams should be approximately 10% higher [6,21,22], which means the value *K_t_*_1_^#^ ≈ 3.1∙1.1 = 3.4. Also, diagrams for the tensile loaded strip with a two-sided reduced height are suitable. From Figure 12.6b [4] and chart 3.1 [5] for the same ratio *H*/*h* and *r*/*h K_t_*_2_* ≈ 2.8 can be obtained and from Figure 3.11 in [6] for the same ratio *r*/*h* = *r*/*t* and the ratio *h*/*H* = *t*/*T* = 0.667 (in our case *h*/*H* = 4.7/6.8 = 0.69), *K_t_*_3_* ≈ 3.0 can be obtained. In the case of a one-sided reduction of the cross-section of the strip, the stress concentrator is approximately 10% lower at a similar ratio *H*/*h* and *r*/*h* [23]. However, since the values from the Peterson’s diagrams should be approximately 10% higher, the final values are *K_t_*_2_^#^ = 2.8 and *K_t_*_3_^#^ = 3.0. The mean value of the stress-concentration factor in the sharp transition between the ribs and the wall is therefore *K_t_*^#^ = (3.4 + 2.8 + 3.0)/3 ≈ 3.1. However, since the crack is initiated in the line where the external diameter and the wall thickness change simultaneously, due to the large rigidity the stress concentrator is also slightly larger. If we assume a 10% larger stress concentration in this area, the final value is *K_t_* = 3.1∙1.1 = 3.4.

Since the crack initiates and grows along a vertical rib, the tangential stress in the cylindrical part of the housing is responsible for the burst and was calculated using Equation (1). By inserting *p* = *p_d_* = 34 bar, *t_w_* = 4.7 mm, *D_in_* = *D_ex_* − 2*t_w_* = 70 − 2·4.7 = 60.6 mm, and *K_t_* = 3.4, the destructive stress of the filter housing with vertical ribs is *σ_t_* = *R_m_* = 74.5 N/mm^2^, which is in the range of the literature values for the strength of SAN.

#### 3.3.2. (b) Sample of Series C

The crack initiation showed that the critical site of the housing without vertical ribs is the central part of its base. The torn off base also shows large stresses at the edges of the base. For the analysis of the stress state it is necessary to know the exact geometrical characteristics of the lower part of the housing. From the cross-section through the base it is evident that on the inside and on the outside the central area of the base is additionally thickened (Figure 1e). The surface of this area on the inside of the housing is convex and its diameter is smaller than the diameter of the outer thickened part. The maximum thickness of the middle of the base is *t_w_* ≈ 7.0 mm and it does not represent a critical point, despite the local surface reduction of the thickness due to a production defect on the outside of the base. The crack was initiated on the outer side of the base at the edge of the thickened part, without the influence of a nearby stress concentrator (Figure 4c and Figure 6c), so *K_t_* = 1. The thickness at the crack-initiation point is *t_w_* = 5.5 mm. Due to the complex shape of the base, the internal diameter was assumed to be *D_in_* = *D_exb_* − 2*t_w_* = 68.6 − 2·4.3 = 60 mm. The calculations show that the base of the housing is fixed. If *D_in_* = 60 mm, *t_w_* = 5.5 mm and *p* = *p_d_* = 50 bar are inserted into Equation (3) for the supported bottom, the stress is too high (*σ_max_* = 184 N/mm^2^) and the burst would be at a much lower pressure. Using Equation (6) for the fixed bottom, the stress in the middle of the base at the initiation site is *σ_r_* = *σ_t_* = 72.6 N/mm^2^ and is consistent with the strength of the SAN.

Why did the crack not initiate on the edge of the base in the curved part, where the maximum stresses exist? The reason is the ring on the outer side of the base (Figure 1d,e), which increases the thickness of the base wall from *t_w_* = 4.3 mm to *t_w_* ≈ 10 mm in the region of maximum stresses. From Equation (4) it follows that at the point of maximum stresses, at the pressure *p_d_* = 50 bar, the actual stress is only *σ_r_* ≈ 34 N/mm^2^. Some maximum stress in the wall thickness *t_w_* = 4.3 mm therefore exists somewhere outside the region of maximum stresses (see the schematic distribution of the longitudinal stresses in Figure 9b). The stress is certainly greater than *σ_r_* ≈ 34 N/mm^2^ and due to the absence of the initial site, it is lower than the strength of the material. So what are the stresses on the edge in the curved part, where at the end of the pressure test, the base is torn off? Since the exact distribution of the meridian stresses in the region of curvature is not known, we assumed that the radial and tangential stresses were very similar in absolute value and then used Equation (5) to estimate the stress. The calculation shows that the tangential stress at the edge, taking into account *D_in_* = 60 mm, *t_w_* = 4.3 mm and *p* = *p_d_* = 50 bar, is *σ_t_* ≈ 55 N/mm^2^, which is 0.75*R_m_*. If we take into account that the stress-intensity factor at the tip of the propagating cracks is maintained, this value of longitudinal stress is definitely sufficient for its stable growth. Since the transverse cracks between the triangular segments were propagated from the internal surface to the external surface of the housing, the stress-intensity factor is further increased by the effect of the water pressure, which further opens the crack. And since the crack along the wall grows much faster than in the depth, at the end of the destructive pressure test, a carrier cross-section in the curved part is small enough that the stress in the curved part exceed the strength of the material. At the same time, the tangential stress in the cylindrical part of the housing without the external ribs is only *σ_t_* ≈ 41 N/mm^2^ (*D_in_* = 70 mm, *t_w_* = 4.3 mm, *p* = *p_d_* = 50 bar and *K_t_* = 1) according to Equation (1) and has no effect on the final bursting of the housing.

#### 3.3.3. (c) Sample of Series D

The crack initiation showed that the critical location of the housing without vertical ribs with the thicker base is a sharp transition between the clamping ring and the wall on the outside of the housing. The sharp transition *r*_2_ = 0.3 mm is a strong elastic stress concentrator *K_t_* and was determined from the same diagrams as in the case of a housing with vertical ribs. From the diagram with the one-sided reduction of the wall of the pressure vessel (chart 3.6 in [5]), it follows that the ratio between the different wall thicknesses is *H*/*h* = *t_wc_*/*t_w_* = 7.2/4.3 ≈ 1.7, while the ratio between the transition radius and the thinner wall is *r*/*h* = *r*_2_/*t_w_* = 0.3/4.3 = 0.07. With an imaginary curve slightly above *H*/*h* = 1.5 we obtain the value *K_t_*_1_* ≈ 2.9. In reality the values from Peterson’s diagrams need to be increased by approximately 10% [6,21,22], so a decisive value is *K_t_*_1_^#^ ≈ 2.9∙1.1 = 3.2. Also, the diagrams for a tensile loaded strip with the two-sided reduced height are suitable. From Figure 12.6b [4] and chart 3.1 [5], with the same ratio *H*/*h* and *r*/*h*, *K_t_*_2_* ≈ 2.45, from Figure 3.11 in [6] for the same ratio *r*/*h* = *r*/*t* and the ratio *h*/*H* = *t*/*T* = 0.667 (in our case *h*/*H* = 4.3/7.2 = 0.6), *K_t_*_3_* ≈ 2.6 can be obtained. In the case of a one-sided cross-section reduction, the stress concentrator is approximately 10% lower for a similar ratio *H*/*h* and *r*/*h* [23]. However, since the values from the Peterson’s diagrams need to be increased by approximately 10%, the final values are *K_t_*_2_^#^ = 2.45 and *K_t_*_3_^#^ = 2.6. The mean value of the stress concentrator in the sharp transition between the clamping ring and the wall is *K_t_*^#^ = *K_t_* = (3.2 + 2.45 + 2.6)/3 ≈ 2.75.

Since the crack is initiated and grows along the clamping ring, the axial stress in the housing is responsible for the burst, which is calculated using Equation (2). By inserting *p* = *p_d_* = 71 bar, *t_w_* = 4.3 mm, *D_in_* = *D_exu_* − 2*t_w_* = 71.6 − 2·4.3 = 63 mm and *K_t_* = 2.75, the destructive stress of the water-filter housing without the vertical ribs is *σ_a_* = *R_m_* = 71.5 N/mm^2^. The value is consistent with the strength of the SAN and is comparable with previously obtained results.

From the results of the calculated destructive stresses it follows that the average tensile strength for the SAN of the water-filter housings is *R_m_* = (74.5 + 72.6 + 71.5)/3 ≈ 73 N/mm^2^. This means that in the water-filter housings the crack was initiated by the formation of crazes at a stress *σ_cr_* ≈ 36.5 N/mm^2^ or at the half value of the destructive pressure *p_cr_* = 0.5 *p_d_* (in series A: *p_cr_* ≈ 17 bar, in series C: *p_cr_* ≈ 25 bar, in series D: *p_cr_* ≈ 35 bar).

### 3.4. Estimation of the Crack-Growth Conditions

The depth of the mirror zone *a_mz_* represents the critical depth of the crack when the cracking of the water-filter housings starts to grow uncontrollably, up until the final break. This means that the stress-intensity factor exceeded the fracture toughness of the SAN. In the case of the crack-initiation on the surface, the fracture toughness *K_IC_* of the brittle polymers can be estimated with Equation [2]:(7)$$K_{IC}=1.12\cdot R_{m}\cdot \sqrt{\pi \cdot a_{mz}}\;(\mathrm{MPa}\;m^{1/2})$$

In the equation, *R_m_* is the strength of the SAN (N/mm^2^) and *a_mz_* is the depth of the mirror zone (m). From the critical depths *a_mz_* (Section 3.2.2) and the mean strength *R_m_* = 73 N/mm^2^, different values for the fracture toughness of the SAN in different water-filter housings exist: series A: *K_IC_* = 1.45 MPa∙m^1/2^, series C: *K_IC_* = 1.77 MPa∙m^1/2^, series D: *K_IC_* = 3.55 MPa∙m^1/2^. The different fracture toughness is not a consequence of the different qualities of the material, but due to the different loading conditions of the housings during the destructive pressure test because of different predominant stresses in individual series of the housings. The results very clearly show that the fracture toughness of the SAN is strongly dependent on the stress-increasing rate in the wall, which in our case is determined by the destructive pressure. The lower is the destructive pressure *p_d_*, the faster is the stress increasing in the wall of the housing, and the smaller is the fracture toughness of the SAN. The results are consistent with the statement in [7] that the *K_IC_* can substantially decrease as the loading rate increases. From the samples of series D the estimated fracture toughness *K_IC_* = 3.55 MPa∙m^1/2^ is the highest value of *K_IC_* for the SAN so far. Since there is an unknown acrylonitrile content in the SAN of the tested housings, some of the known data for the fracture toughness are given. For the SAN with a content of 30 wt.% of acrylonitrile the highest value *K_IC_* ≈ 2.1 MPa∙m^1/2^, and with a content of 70 wt.% of acrylonitrile the highest value *K_IC_* ≈ 2.5 MPa∙m^1/2^ is given in [24]. For the same content of acrylonitrile the maximum values *K_IC_* ≈ 2.3 MPa∙m^1/2^ and *K_IC_* ≈ 2.7 MPa∙m^1/2^ are predicted in [2]. It is also important to know that the most common type of SAN is that with 25–30 wt.% acrylonitrile.

The different fracture toughness of the SAN is also clearly evident in the texture of the fracture surfaces. SAN is a brittle material and for a brittle fracture the plane-strain state is responsible. This can be created at a critical thickness *t*, which is determined [2,4,7,10,25,26]:(8)$$t\geq 2.5\cdot {\left(K_{IC}/R_{y}\right)}^{2}\;(m)$$

In the equation, *K_IC_* is the fracture toughness (MPa∙m^1/2^), *R_y_* is the yield strength (N/mm^2^), and 2.5 is a commonly used geometrical constant. For the polymers, the values of the geometrical constant were experimentally determined over a very wide range from 0.24 to 26 [27], but the general known value is sufficient for SAN [2]. Because of the fact that *R_y_* ≈ *R_m_*, for brittle polymers, Equation (8) can be written as:(9)$$t\geq 2.5\cdot {\left(K_{IC}/R_{m}\right)}^{2}\;(m)$$

If the previously calculated fracture toughness and the mean strength *R_m_* = 73 N/mm^2^ are inserted into Equation (9), the brittle fracture occurs at the wall thicknesses *t* ≈ 1.0 mm (series A), *t* ≈ 1.5 mm (series C) and *t* ≈ 5.9 mm (series D), which is consistent with the measured values (Figure 8a–c). The results show that the critical thickness for the plane-strain state of the SAN is *t* = 10*a_mz_* and it looks like a general relation for the SAN. In the housings of series A and C, due to *t_w_* > *t*, the rapidly growing cracks grow first in the area of the plane-strain state (here the hackle lines are curved), and finally, in the plane-stress state (here, the Wallner lines are approximately parallel to the external surface, while the hackle lines are approximately perpendicular to it; Figure 8a,b). In the samples of series D due to *t_w_* < *t*, the cracks grow all the time in the plane-stress state with a strong shear plastic tearing of the material (Figure 8d). Macroscopically, these cracks look like a pine twig or feathers due to the densely distributed, short secondary cracks that grow from the main cracks (Figure 5c–e). From the transformed Equation (9) it is evident that in a water-filter housing with a wall thickness of *t_w_* = 4.3 mm, and a SAN strength of *R_m_* = 73 N/mm^2^, the crack grows in a plane-stress state at a *K_IC_* >3.0 MPa∙m^1/2^.

From the analysis of the crack-growth direction, macroscopic and microscopic analyses of the fracture surfaces and from various *K_IC_* values, it follows that the fracture toughness of the SAN is already changing in one sample during the destructive pressure test due to the changeable growth rate of the various cracks. The reason is the constant increasing of the pressure during the destructive pressure test and the different dominant stresses acting in the different housings. The fact is that if the crack surface is as smooth and transparent as possible (this excludes an initial area with a sub-critical crack that is also macroscopically very smooth) it means that the faster is its growing rate, the lower is the fracture toughness of the SAN, and the smaller is the critical thickness for the formation of the plane-strain condition.

### 3.5. Summary of the Results

Table 1 presents a comparison of the practically and theoretically obtained results for all three different series of tested SAN water-filter housings. The results show the effect of different critical areas on the formation of different dominant stresses and the effect of dominant stresses on the magnitude of the destructive pressure, fracture toughness, depth of the subcritical crack (mirror zone) and the critical wall thickness for the brittle fracture. It is clear that with the increasing destructive pressure, all the fracture-mechanical quantities increased, which is a consequence of the slower rise of the stress in the housing wall and the slower crack propagation. Because of that, with an increasing destructive pressure, the state of the crack propagation changes from the plane-strain state to the plane-stress state.

## 4. Conclusions

For the same volume, but with differently formed cylindrical SAN water-filter housings, a destructive water-pressure test and an analysis of the fracture surface were experimentally performed. Also, the theoretically stress condition in the walls of the housings, the strength of the SAN and the crack-initiation and propagation conditions were assessed. Based on the results, the following can be concluded:-The tensile strength of the SAN of the water-filter housings is *R_m_* = 73 N/mm^2^, which is in the range of the literature data. In spite of the same tensile strength, the shape of the housings with its details strongly affects their critical area and the value of the destructive pressure. The least destructive pressure *p_d_* = 34 bar was for the housing with external vertical ribs. The crack initiated at a sharp transition between the rib and the wall with *K_t_* ≈ 3.4, as a result of the tangential stress *σ_t_*. The largest destructive pressure *p_d_* = 71 bar was for the housing without the ribs with the thickened base. Here, the crack initiated at a sharp transition between the clamping ring and the wall with *K_t_* ≈ 2.75, as a result of the axial stress *σ_a_*. Between these two examples is a housing without the ribs with a uniform thick wall and a destructive pressure of *p_d_* = 50 bar. Here, the crack initiated in the middle of the thickened base without a stress concentrator as a result of the stress *σ_r_* = *σ_t_*.-The different shape of the housings with associated various critical sites and the various destructive pressures strongly affects the initiation areas, the direction, the kinetics, the mode of growth and the splitting of the cracks during their propagation. They represent unique and repeatable parameters for each type of housing individually.-The critical depth of the cracks, the fracture toughness of the SAN, and the critical thickness of the wall for the formation of a brittle fracture are very dependent on the stress-increasing rate or the velocity of the crack propagation. The lower the destructive pressure is, the faster is the stress-increasing rate and the smaller is the critical length of the crack *a_mz_* required for the formation of uncontrolled crack growth. Furthermore, the crack growth is also faster. Consequently, at a lower destructive pressure, the fracture toughness of the SAN is lower as well as the critical thickness of the wall for the formation of a brittle fracture.-The SAN from which the water-filter housings were made has the following fracture-mechanical properties at destructive pressures *p_d_* = 34, 50 and 71 bar, respectively: fracture toughness *K_IC_* ≈ 1.45, 1.77 and 3.55 MPa∙m^1/2^; depth of the mirror zone *a_mz_* ≈ 100, 150 and 600 μm; the critical thickness for the formation of a brittle fracture *t* ≈ 1.0, 1.5 and 6.0 mm or *t* = 10*a_mz_*. Therefore, the crack in the wall of the samples of series A and C grows predominantly in the plane-strain state, while the crack in the sample of series D grows in the plane-stress state.-The crack-growth velocity affects the surface texture of the mirror zone. In all cases, regardless of the growth velocity of the sub-critical cracks, the mirror zone is microscopically very finely rough as a result of the formation and tearing of the crazes. On the surface of the mirror zone of the slowest-growing sub-critical crack, different shape of contours also exist, and the most typical of this is a large number of parabolic markings. These are oriented with an open side in the growing direction of the main crack. All the observed contours are the result of the initiation of secondary micro-cracks at micro-unhomogeneities sites in the SAN, which intersect their growth fronts with the main crack. This is only the case for a sufficiently small growth rate of the main crack, when the micro-unhomogeneities can act as a local stress micro-concentrator ahead of the growing front of the main crack. If the crack-growth velocity of the main crack is greater than the critical cracking rate for this phenomenon, the surface of the mirror zone is finely rough without the mentioned contours.-The texture of all the analyzed over-critical surface cracks in the vicinity of the mirror zone is same due to the initial small growth rate, regardless of the destructive pressure. Macroscopic and microscopic semi-elliptical Wallner lines and from the initial area divergent-oriented hackle lines are visible. The Wallner lines generally show the front of the growth crack, and the hackle lines are directed towards the direction of the main crack growth.-The crack-growth velocity affects the surface texture of the over-critical unstable cracks. In the subsequent rapid-growing over-critical cracks in the plane-strain state, the Wallner lines and hackle lines are strongly curved, indicating a different cracking velocity through the wall thickness. The Wallner lines and hackle lines together form a group of intersecting curves that are mathematically described as a system of a single-parametric family of curves with a family of orthogonal trajectories. On the surface of the slowly growing over-critical crack in the plane-stress state at some distance from the initiation area a strong plastic shear tearing of the material with a lot of secondary micro-cracks, initiated at micro-unhomogeneities in the styrene–acrylonitrile are visible. Macroscopically, such cracks appear in the form of a pine twig or feathers, and microscopically on the fracture surface they have a lot of rosette-like forms. On a fracture surface, far enough away from the initial area, only the hackle lines exist. This is a consequence of a very rough and split crack-growing front that causes a strong, diffuse reflection of the returned shock waves, and which prevents the formation of the Wallner lines. The proof of this is the beginning slow-growing phase of the over-critical crack near the initial area. There the Wallner lines are clearly visible, despite the low crack-propagation velocity due to the smooth crack surface, which is a consequence of the non-split crack-growing front.

## Figures and Tables

**Figure 1 polymers-12-00280-f001:**
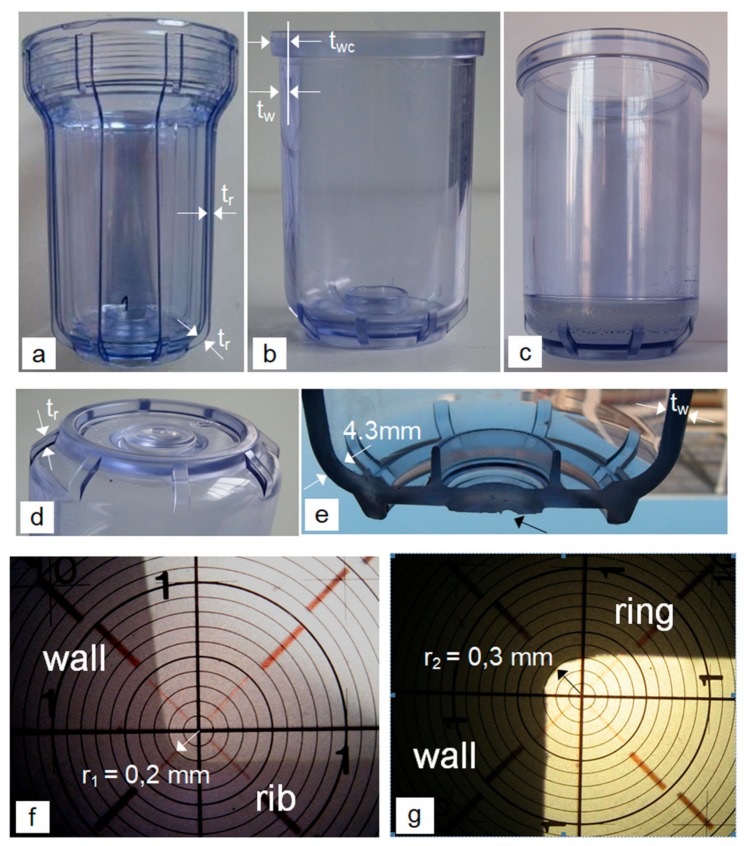
Types of water-filter housings and some of their geometric characteristics (**a**) housing with the external vertical ribs [1] (**b**) housing without the vertical ribs (**c**) housing without the vertical ribs with an increased base thickness (**d**) the base of the housing without the vertical ribs (**e**) the cross-section of the base between the ribs (**f**) the transition radius *r*_1_ between the ribs and the wall (**g**) the transition radius *r*_2_ between the clamp ring and the wall.

**Figure 2 polymers-12-00280-f002:**
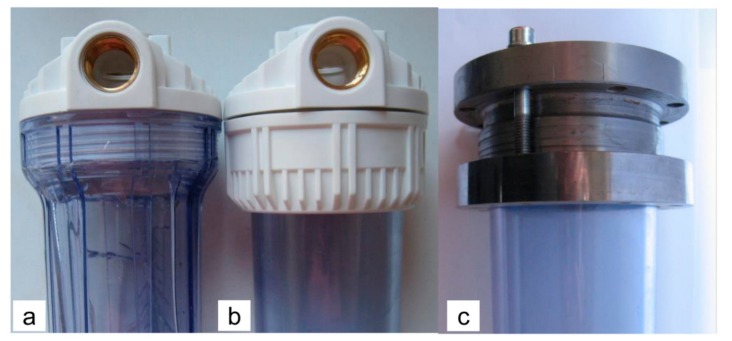
Different ways of fitting the housings to the head for the different samples (**a**) series A (**b**) series B (**c**) series C and D.

**Figure 3 polymers-12-00280-f003:**
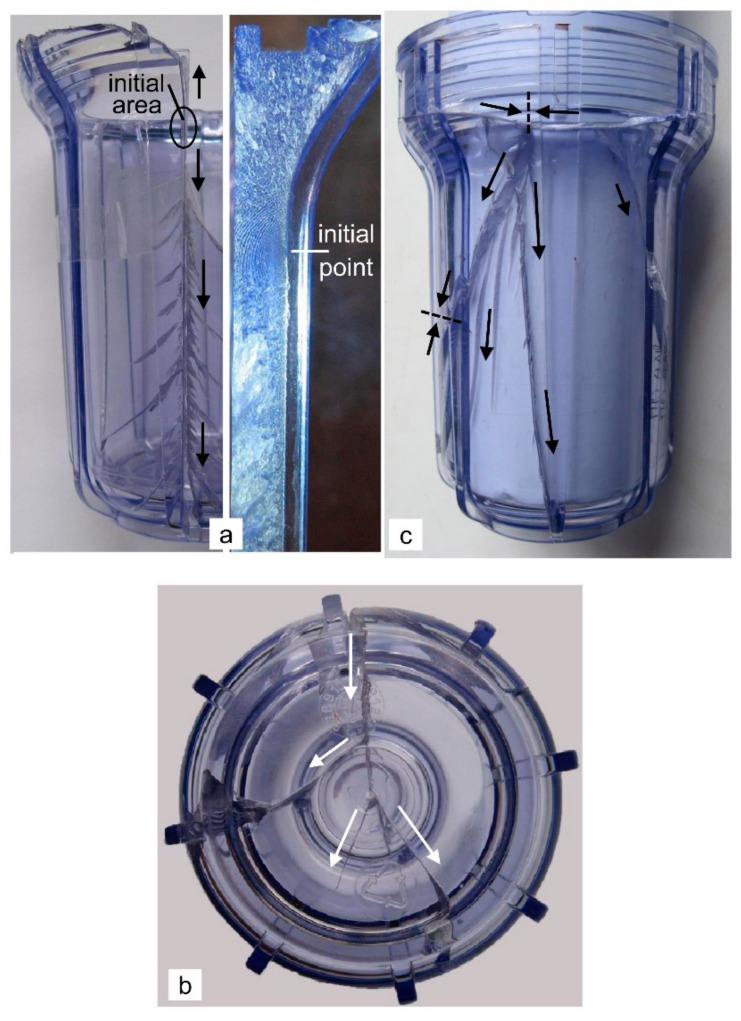
Specific characteristics of the initiation and propagation of the cracks in the sample of series A (**a**) the location of the initiation and downwards growth of the primary crack at the rib [2] (**b**) branching the primary crack in the middle of the base [2] (**c**) the junction site of the two oppositely growing horizontal cracks and the branching of the newly formed downwards-growing crack on the rear side of the filter housing [2]; the arrows indicate the direction of the crack growth.

**Figure 4 polymers-12-00280-f004:**
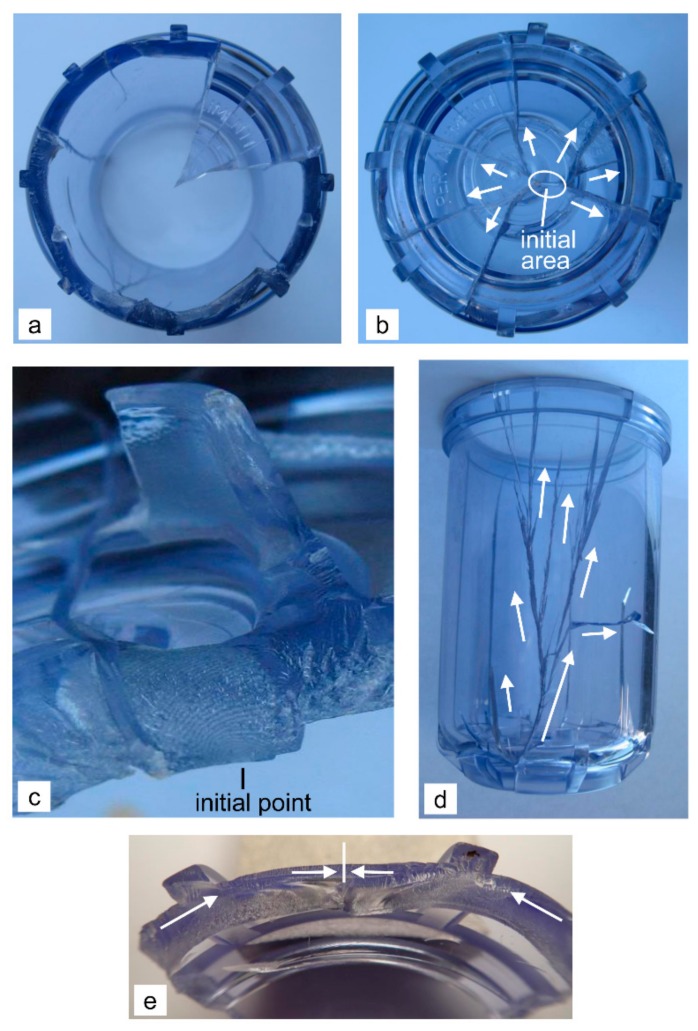
Specific characteristics of the initiation and propagation of the cracks in the sample of series C (**a**) the breaking base (**b**) an initiation area in the central thickened cylindrical part of the base and radial spreading of the cracks through the base (**c**) an initiation point on the external side of the thickened cylindrical part (**d**) the branched upwards-growing crack (**e**) the oppositely growing horizontal cracks joining at the middle of the fracture in the arc of the circular sector’s fragment; the arrows indicate the direction of the crack growth.

**Figure 5 polymers-12-00280-f005:**
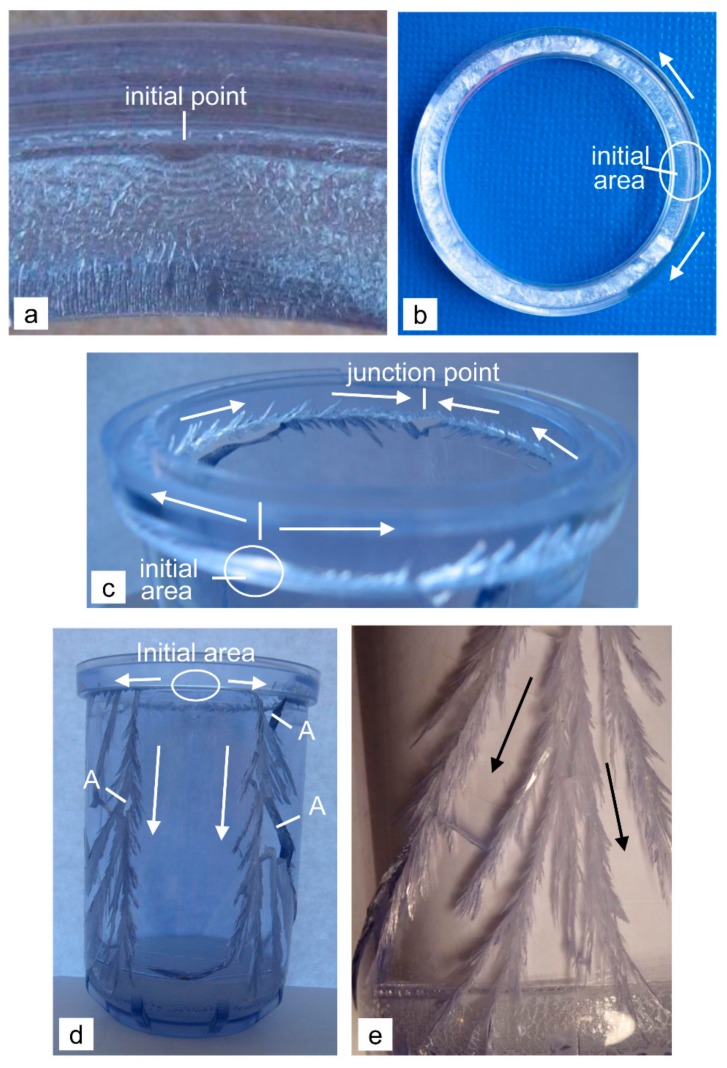
Specific characteristics of the initiation and propagation of the cracks in the sample of series D (**a**) an initiation point (**b**) a detached clamping ring (**c**) junction of the oppositely growing horizontal cracks with a small detached triangular fragment (**d**) strong splitting of vertically downwards-growing cracks (**e**) very densely arranged short secondary cracks along the main cracks; the arrows indicate the direction of the crack growth.

**Figure 6 polymers-12-00280-f006:**
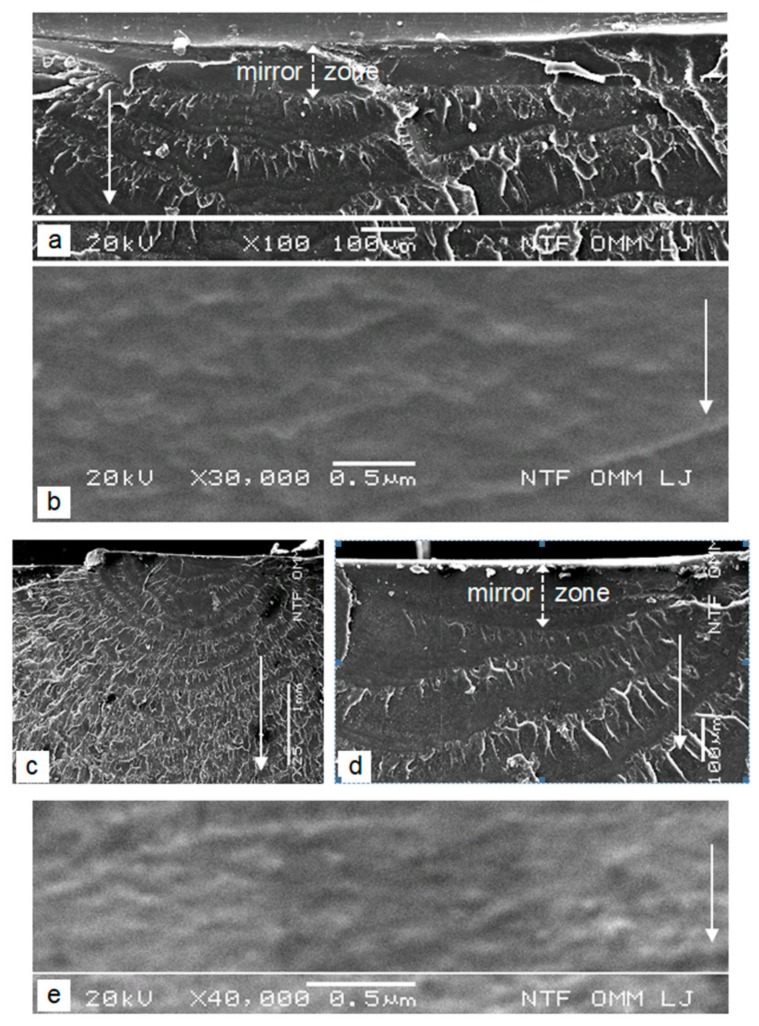
Characteristics of the mirror zones (**a**,**b**) sample of series A [2] (**c**–**e**) sample of series C; the solid arrows indicate the direction of the main crack growth.

**Figure 7 polymers-12-00280-f007:**
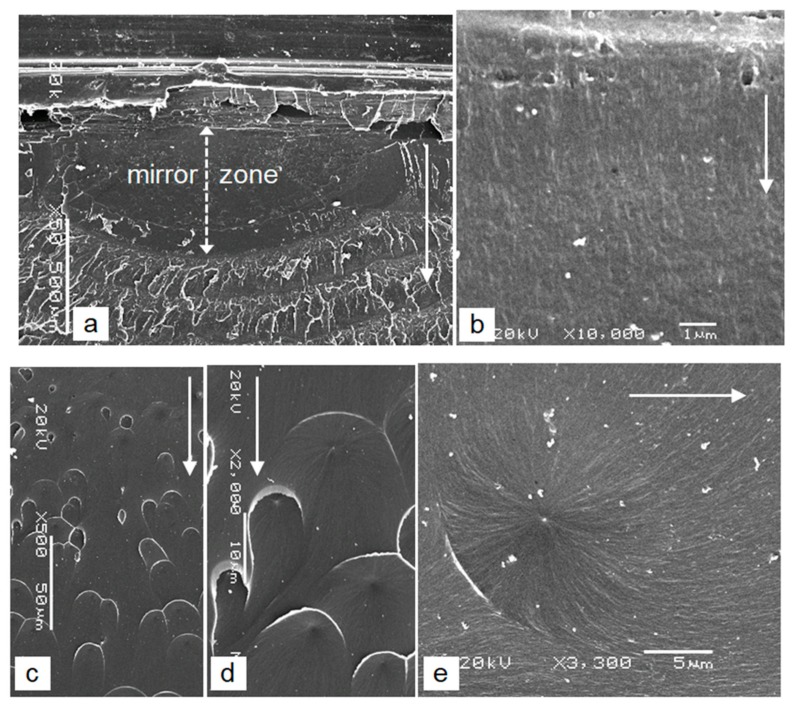
Characteristics of the mirror zone of the sample of series D (**a**) symmetric semi-elliptic form of the mirror zone (**b**) lines in the direction of the main crack (**c**) different tips of contours in the crack surface of the mirror zone (**d**) parabolic markings with the initials in their focus (**e**) micro-crater with the initial in its center; the solid arrows indicate the direction of the main crack growth.

**Figure 8 polymers-12-00280-f008:**
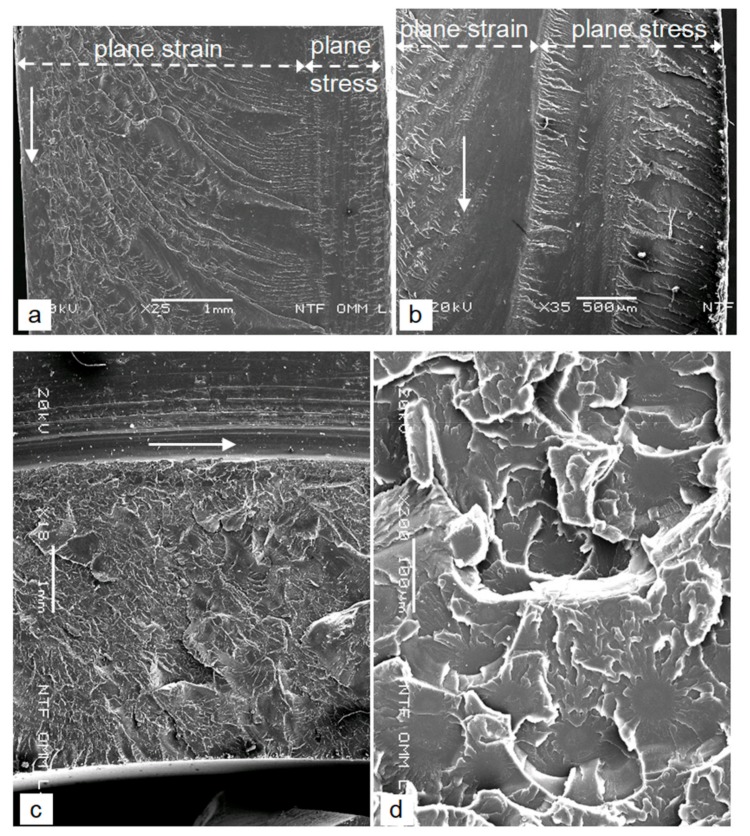
Crack surface texture of the area that is far away from the initial sites (**a**) curved hackle lines in the field of very rapid crack growth and the growth transition from the plane-strain to the plane-stress condition, the sample of series A [2] (**b**) the growth transition from the plane-strain to the plane-stress condition, horizontal crack in the curved part of the bottom of the sample of series C (**c**) slowly growing crack of the detached clamping ring, the sample of series D (**d**) the mixed brittle-ductile character of the crack surface with strong shear plastic tearing, the sample of series D; the solid arrows indicate the direction of the main crack growth.

**Figure 9 polymers-12-00280-f009:**
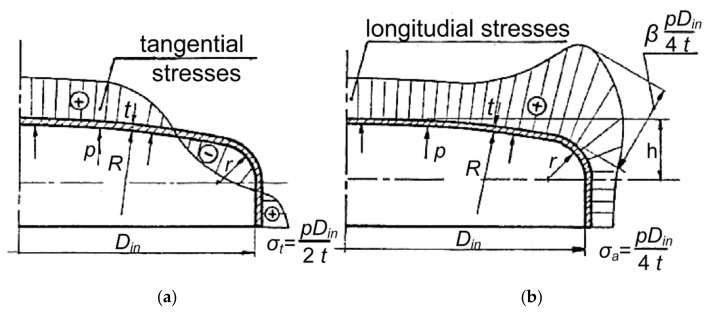
Stresses in the wall of the bottom due to the internal pressure in the vessel (**a**) tangential stresses (**b**) axial stresses [8].

**Table 1 polymers-12-00280-t001:** Geometrical, strength and fracture-mechanical characteristics of destructive pressure-tested water-filter housings.

Sample	*t_w_* (mm)	*K_t_*	Initial Dominant Stress	*p_d_* (bar)	*K_IC_* (MPa⋅m^1/2^)	*a_mz_* (mm)	*t* (mm)	Crack Growth State
Series A	4.7	3.4	*σ_t_*	34	1.45	0.10	1.0	plane-strain
Series C	4.3	1.0	*σ_t_ = σ_r_*	50	1.77	0.15	1.5	plane-strain
Series D	4.3	2.75	*σ_a_*	71	3.55	0.60	5.9	plane-stress

Series A—housing with vertical ribs; Series C—housing without the ribs; Series D—housing without the ribs but with thicker base; *t_w_*—thickness of the wall; *K_t_*—elastic stress concentrator factor; *p_d_*—destructive pressure; *σ_t_*—tangential stress; *σ_r_*—radial stress; *σ_a_*—axial stress; *K_IC_*—fracture toughness; *a_mz_*—mirror zone depth; *t*—critical thickness for the brittle fracture.

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
