# Peer review of "Effect of the Shape of Styrene–Acrylonitrile Water-Filter Housings on the Destructive Pressure, Crack-Initiation, Propagation Conditions and Fracture Toughness of Styrene–Acrylonitrile"

_polymers, 2020, doi:10.3390/polym12020280_

Round 1

Reviewer 1 Report

The authors studied the effect of the shape of styrene-acrylonitrile water-filter housings on the destructive pressure, crack-initiation, propagation conditions and fracture toughness of styrene-acrylonitrile. And the theoretically stress condition in the walls of the housings, the strength of the SAN and the crack-initiation and propagation conditions were assessed. However, the manuscript should be revised before publication. The following are the detailed comments:

1.The “Experimental section”, including materials, instruments for preparing samples and for testing, testing standard, should be added in manuscript. Authors should give some details of SAN (for example, molecule weight or viscosity, molar ratio of styrene in SAN)

2.Section 3.1Geometrical characteristics of the water-filter housings , 3.2 Visual inspection of the water-filter housings , and 3.3 Destructive pressure test should be some parts of “Experimental section”, and should not be in “Result and discussion” section. The shape and size of the ribs in housing samples should be explained in detail in “Experimental section”.

3.The housings (Figure 1.a,b,c) have the same internal diameter, and they also have the same external diameter? If the external diameter are different, what happened to those housing samples?

4.If PP nut on the filter head and epoxy resin of the increased bottom (Fig 1 c) influence the properties of SAN during external press?

5.“Conclusions” section should be more concise by summarizing.

Author Response

Responses to reviewer comments:

We would like to thank you for your remarks and suggestions. We are glad that you didn´t put any scientific and professional comment.

The paper was proofread by English native speaker dr. Paul McGuiness. The Certificate is attached (see attachement).

According to reliever’s comments we have done the following changes in the manuscript:

Ad 1 and Ad 2:

The text in the manuscript was reconstructed in accordance with reviewer suggestions. The new chapter 2. Experimental was added. The most of the text from the Results and discussion was moved into chapter 2. Experimental.

Since the water-filter housings have been bought on the market the information about the chemical and physical properties are not known. Anyway, the article is focused on the mechanical properties of the housings.

Ad 3:

The diameters of the housings are practically the same. The difference in the external diameters is less than 1 mm and it does not have any effects on the results of the analysis in the article. In general, all calculations are based on internal diameter.

Ad 4:

We are not sure if we understand the question correctly. However, the different ways of fitting and artificially increased of base thickness have effect on initial crack area and subsequently on destructive pressure.

Ad 5:

Some changes were made in Conclusions. However, after reading the Conclusions many times, we think that these conclusions show the key points and findings. Moreover, they even bring short explanations. Since other reviewer didn´t have any complaint about the conclusions we would be grateful if we do not need to change them anymore, because we do not have an idea how to write them in different form.

All modifications in the manuscript are highlighted in green and red (highlighted version). Green text was reconstructed or newly added in previous text. Red text will be removed from previous text.

We would like to thank you for your remarks and suggestions which have improved the paper quality.

Hoping that the above mentioned changes in the manuscript and our answers satisfy all reviewer comments, I look forward to hearing from you soon.

Yours sincerely,

Assist. prof. Borut Zorc

Reviewer 2 Report

Title:

Effect of the shape of styrene-acrylonitrile water-filter housings on the destructive pressure, crack-initiation, propagation conditions and fracture toughness of styrene-acrylonitrile

Authors: Borut Zorc and colleagues

Overall assessment:

The authors conducted several researches on the fracturing properties in several water-filter housings with difficult configurations, and they obtained several interesting results from the experiment.

The topic adopted here seems to fascinate several researchers. Even if so, however, the construction of the manuscript is too bad, and the descriptions are extremely casual and ambiguous. To continue the reviewing process, the authors must revise the manuscript thoroughly while considering the issues described in the following comments:

Specific comments:

The construction of the manuscript is very bad; therefore, the authors must revise the construction of the manuscript thoroughly. For example, the details on the experimental procedures must be denoted in Section 2, and only the corresponding results must be denoted in Section 3. In the present status, the descriptions in Section 2 are too terse.

When seeing Fig. 1 alone, it is difficult to understand the portions to which the values r1, r2, tw, tr, and twc correspond. In particular, it is quite difficult to understand what Figs. 1(f) and (g) represent. I strongly recommend the authors to provide a diagram of the typical housing for describing these values.

In the experiment, only two housings were used corresponding to each test condition. However, the sample numbers must be increased, mustn’t they? In the tests conducted by the authors, it is difficult to reduce any anomalous data, and it is impossible to discuss the validity of the tests in statistical aspect.

If the authors think that the sample numbers are valid for the test, they must describe the validity in the manuscript.

Line 141: “bar” is missing.

The descriptions in Subsections 3.4.1 and 3.4.2 are extremely tedious to read. Although the visual observations are important for this study, they are often dominated from the subjectivities of the authors. The objective values later described are more important. Therefore, I strongly recommend the authors to elaborate the descriptions.

I recommend the authors to demonstrate Subsection 3.4.2 using the style similar to Subsection 3.4.1, which is arranged according to each sample.

There are solid and broken arrows in Figs. 6-8. The difference of these arrows must be denoted in the figure caption.

The descriptions in Subsections 3.5 and 3.6 are also tedious to read. In the present descriptions, many properties obtained here are arbitrarily demonstrated; therefore, it is difficult for many readers to appreciate the importance of the study at all. The authors must elaborate the descriptions thoroughly into those many readers can understand the essence more easily and directly.

Why the authors do not use any tables for demonstrating the values of the properties obtained in this study? I think some of tables will aid the clear demonstrations of the manuscript.

Recommendation

Thorough revisions are required and the authors must provide a point-by-point response to the comments. Even if the topic is interesting for many readers, bad descriptions prevent the readers from the precise understandings of the essence. Considering the convenience for the readers, the authors must revise the manuscript with careful and objective readings. I’d like to wait for receiving the revised version in due course.

Author Response

Responses to reviewer´s comments:

We would like to thank you for your remarks and suggestions. We are glad that you didn´t put any scientific and professional comments.

The paper was proofread by English native speaker dr. Paul McGuiness. The certificated is attached (see attachement).

According to reliever’s comments we have done the following changes in the manuscript:

The text in the manuscript was reconstructed in accordance with reviewer´s suggestions. The new chapter 2. Experimental was added. The most of the text from the Results and discussion was moved into chapter 2. Experimental.

Figure 1 was modified. New marks for r1, r2 tw, tr and twc were added for the clarification. Figures 1f and 1g present the evidence that r1 is really 0.2 mm and r3 is really 0.3 mm because these are important for calculation of the stress concentrator factor Kt.

Practical results confirm that two samples from each series are sufficient for the credibility of the results. No anomalies were found in our results which is now explained in last paragraph of the chapter 3.2 Analysis of the fractures.

˝Bar˝ has been added in previous line 141 (now this line is in 3.1 Destructive pressure test).

Subsections 3.4.1 and 3.4.2 are now 3.2.1 and 3.2.2. The visual examination for such an experiments are very important because macroscopic and microscopic characteristic give all information about crack initiating and propagating. Besides the imaging the visual description is the only way for the illustration of the phenomenon. Namely, without precise visual analysis the calculated results of fracture mechanical parameters wouldn´t be credible because they wouldn´t be confirmed. However, the way of writing has subjective character and thus, the text of the chapters hasn´t been changed significantly. We should note that the other reviewer didn´t have any complaints about reading this text tediously.

Subsection 3.4.2. (now 3.2.2) deals with scanning electron microscopy. Since the results of series A and C are practically the same, the proposed reconstruction of the text would result in repeating of the same text. Therefore, the text in this subsection is written according to two different state on the fracture surface.

On figures 6, 7 and 8 the solid arrows represent the crack growth direction which is explained in figure captions. While the dashed lines represent quotation of geometrical characteristics which can be clearly seen from the figures.

We have partly reconstructed the article and added some new text to improve the quality of the paper. We have also added Table 1 as you proposed. Anyway, we would like to emphasize that the other reviewer didn´t have any complaints about reading the text tediously. This indicates that the way of writing has subjective character and it is difficult to satisfy every reader. However, both reviewers had the same remarks about text in section 2 and 3 which has now been reconstructed. In our opinion the new version of the paper is a good compromise between comments of both reviewers.

All modifications in the manuscript are highlighted in green and red (highlighted version). Green text was reconstructed or newly added in previous text. Red text will be removed from previous text.

Hoping that the above mentioned changes in the manuscript and our answers satisfy all reviewer comments, I look forward to hearing from you soon.

Yours sincerely,

Assist. prof. Borut Zorc

Round 2

Reviewer 2 Report

Several minor amendments are adequately conducted. I still think that the manuscript is totally tedious and lengthy, I also think that the readers will determine the value of the manuscript; therefore, I'd like to recommend the revised version to be published as it is.